# The First Attempt of Satellite Tracking on Occurrence and Migration of Bryde's Whale (*Balaenoptera edeni*) in the Beibu Gulf

**Mingming Liu** [1], **Wenzhi Lin** [1], **Mingli Lin** [1], **Binshuai Liu** [1,2], **Lijun Dong** [1], **Peijun Zhang** [1], **Zixin Yang** [1], **Kexiong Wang** [3], **Liang Dai** [1] and **Songhai Li** [1,4,*]

[1] Marine Mammal and Marine Bioacoustics Laboratory, Institute of Deep-Sea Science and Engineering, Chinese Academy of Sciences, Sanya 572000, China; liuming@idsse.ac.cn (M.L.); linwz@idsse.ac.cn (W.L.); mingli@idsse.ac.cn (M.L.); liubs@idsse.ac.cn (B.L.); donglj@idsse.ac.cn (L.D.); pjzhang@idsse.ac.cn (P.Z.); yangzx@idsse.ac.cn (Z.Y.); dai@idsse.ac.cn (L.D.)
[2] College of Marine Sciences, University of Chinese Academy of Sciences, Qingdao 266400, China
[3] Key Laboratory of Aquatic Biodiversity and Conservation, Institute of Hydrobiology, Chinese Academy of Sciences, Wuhan 430072, China; wangk@ihb.ac.cn
[4] Center for Ocean Mega-Science, Chinese Academy of Sciences, Qingdao 266071, China
[*] Correspondence: lish@idsse.ac.cn

**Abstract:** Satellite-tagging is increasingly becoming a powerful biotelemetry approach to obtain remote measurement through tracking free-living cetaceans, which can fill knowledge gaps on cetaceans and facilitate conservation management. Here, we made a first biologging attempt on baleen whales in Chinese waters. An adult Bryde's whale in the Beibu Gulf was tagged to investigate potential occurrence areas and migration routes of this poorly studied species. The whale was satellite-tracked for ~6 days with 71 filtered Argos satellite locations, resulting in a linear movement distance of 464 km. At each satellite-tracking location, the water depth was measured as 42.1 ± 24.8 m on average. During the satellite-tracking period, the whale's moving speed was estimated at 5.33 ± 4.01 km/h. These findings expanded the known distribution areas of Bryde's whales in the Beibu Gulf and provided an important scientific basis for the regional protection of this species. We suggest that fine-scale movements, habitat use, and migratory behavior of Bryde's whales in the Beibu Gulf need more biotelemetry research, using long-term satellite-tracking tags and involving enough individuals. Furthermore, the genetic relationship and possible connectivity of Bryde's whales in the Beibu Gulf and adjacent waters should be examined.

**Keywords:** biotelemetry; biologging; satellite tag; movement; migratory behavior; conservation management

## 1. Introduction

Effective conservation management of threatened cetacean species requires robust scientific data on cetacean behavior and ecology [1], which typically come from boat-based field investigations. However, many cetacean species often travel great distances and occur in remote environments, which present numerous challenges for researchers to conduct field investigations [2,3]. Over the last several decades, biotelemetry (e.g., radio telemetry, acoustic telemetry, satellite tracking) is increasingly being applied in research on animal ecology and behavior in the wild [4,5], which can be extremely helpful in filling knowledge gaps in movement, habitat use, physiology, and/or acoustic behavior of highly mobile cetaceans [6]. Among various technologies of biotelemetry, satellite tagging represents a powerful biologging approach to obtain remote measurements of occurrence locations, diving depth, seawater temperature, and/or other available parameters by tracking free-living cetaceans [4–7].

Most baleen whale species have a general pattern of migration behavior toward the equator in winter and toward higher latitude waters in summer [3]. However, the Bryde's whale *Balaenoptera edeni* (Anderson, 1879) can be observed all year round in tropical, subtropical, and warm temperate waters of the Pacific, Indian, and Atlantic Oceans, typically between 40° N and 40° S [8]. As one of the least known baleen whale species, the Bryde's whale includes a complex set of several subspecies and possible species [9,10]. Currently, it is recognized with two provisional subspecies: "small and coastal form"—*B. e. edeni*, and "large and oceanic form"—*B. e. brydei* [11,12]. The Bryde's whale was globally classified as "Least Concern" by the Red List of Threatened Species, International Union for Conservation of Nature [13], but a few geographical populations were regionally assessed "Endangered" or "Critically Endangered" [14,15]. Nevertheless, there are numerous data-deficient areas with scant field investigation on this species [16]. Especially, fundamental information on the occurrence of this species in many developing countries and regions is often anecdotal, outdated, or even nonexistent, leading to considerable knowledge gaps and challenges in the regional and international protection of this species [8,13,16].

Historic whaling activities and sporadic stranding records indicated that the Bryde's whale could be found in the coastal waters of many countries around the South China Sea (Figure 1A), including China [17,18], Thailand [19,20], Vietnam [21], Malaysia [22], Singapore [23], Philippines [24], and Indonesia [25]. However, these records usually confused Bryde's whales with Sei whale *B. borealis* and/or Omura's whale *B. omurai* [8,9,17]. Thus far, there have been at least two confirmed hotspot waters with the regular occurrence of Bryde's whales in the South China Sea: one in the northern Beibu Gulf [26,27], and the other in the upper Gulf of Thailand [19,20,28] (Figure 1A). According to morphologic and oceanographic characteristics, these two populations were believed to be "small and coastal form"—*B. e. edeni*, as the body length of whales in these two gulfs were typically estimated less than 10 m, and these two gulfs are shallow seas far from deep and offshore waters [20,26,27].

The fine-scale occurrence of Bryde's whales near the Weizhou Island in the northern Beibu Gulf, China was first reported by social media videos in 2015 [26]. Subsequent studies documented data on Bryde's whale encounters and tread-water feeding observations in a small area around Weizhou Island between March and April of 2018 [26,27,29]. These authors pointed out that the waters around Weizhou Island might be a feeding ground for Bryde's whales in the northern Beibu Gulf (Figure 1B). However, little information, if anything at all, has been obtained to reveal the movement, habitat use, and potential migration of Bryde's whales in this region. Except for the waters around Weizhou Island, potential occurrence areas of Bryde's whales in the Beibu Gulf have still not been scientifically described.

In this study, we had a preliminary attempt to deploy the anchored satellite tag on a free-living whale in a hotspot area of Bryde's whales in the Beibu Gulf during a local whale-watching season. We tracked the whale by Argos-based satellite-tracking technique in order to investigate potential occurrence areas and migration routes of Bryde's whales. This study aims to offer an important scientific basis for the protection of Bryde's whales in the Beibu Gulf and to provide insights into future research on the whales in this region.

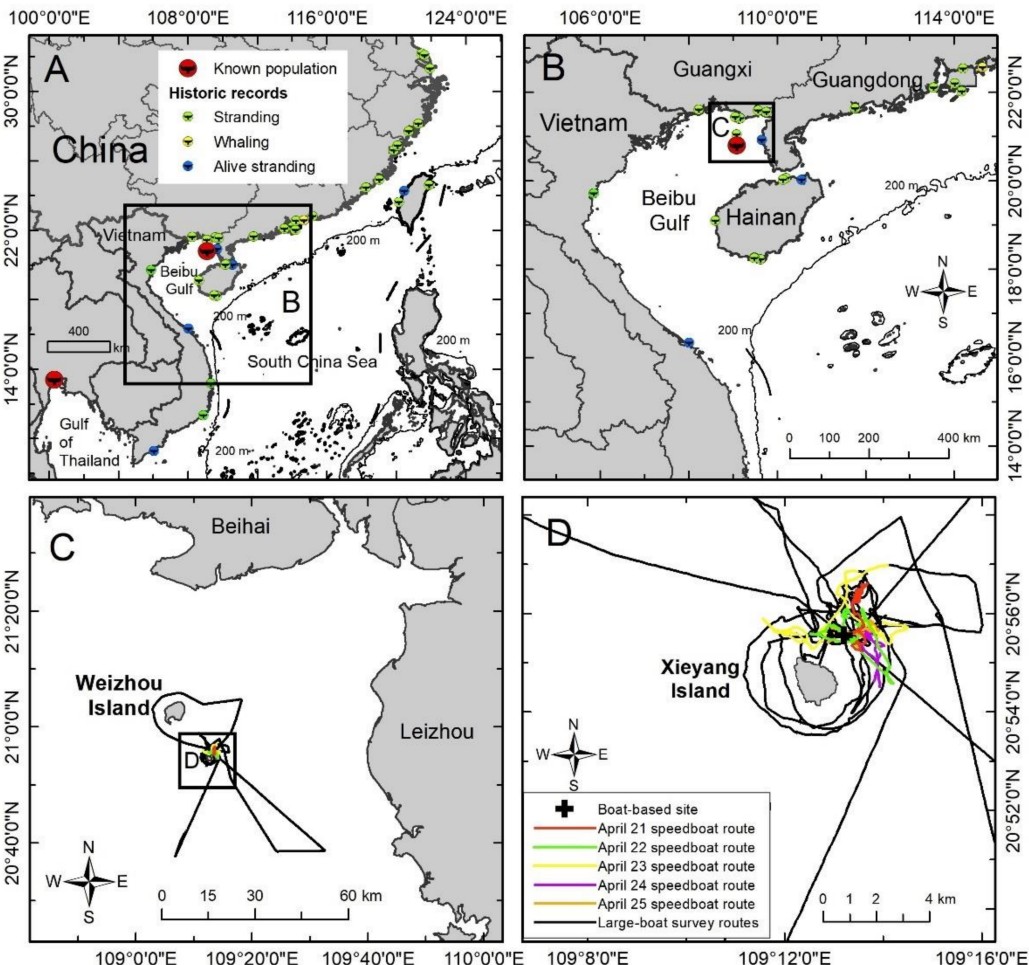

**Figure 1.** Map of the study area: (**A**) the South China Sea, (**B**) the northern Beibu Gulf, (**C**) the waters and boat-based survey routes around Weizhou Island, and (**D**) the waters and speedboat survey routes nearby Xieyang Island, China.

## 2. Materials and Methods

From 21 to 25 April 2021, a tagging project was conducted for Bryde's whales in the waters around Weizhou Island, Beibu Gulf, China (Figure 1B). We used a 40 m length large fishing vessel (450 gross tons, cruising speed: ~6–8 knots) and an 8.8 m length speedboat (115 HP four-stroke engine, maximum speed: ~30 knots) to search Bryde's whales in the study area (Figure 1C,D). Furthermore, a fixed location (20.93° N, 109.22° E) at the waters north of Xieyang Island was chosen for fixed-point visual observation (Figure 1D). The selection of location was based on prior knowledge, as this location was in the center of a hotspot area of Bryde's whales in the Beibu Gulf, especially during the spring whale-watching season [26,27,29]. On the top floor of the boat (~5.6-m above sea level), a platform was established for visual observation [30]. Four primary observers (POs) equipped with 4 handheld 7 × 50 binoculars and one independent experienced observer (IO) with "Bigeyes" 25 × 125 binoculars were on duty to keep searching the whales through observing the 360° of sea surface around the visual platform during the daytime (typically 08:00–18:00, Beijing Time, UTC+08). All observers took 1 h observation on duty and then had a 2 h rest to avoid fatigue in turn [30].

The species identification of Bryde's whales was based on their morphologic and behavioral characteristics [8,17,26–28]. In addition, we recorded information on the date, time, location, group size (number of estimated individuals) [30], whether tread-water feeding activities were present [27,28], whether mother–calf pair was present (if yes, number of pairs) [20], and whether whale-watching boats were present (if yes, number of boat trips)



on a data sheet. Whenever possible, the small speedboat was used to conduct focal follows for any opportunities of tagging Bryde's whales. In total, we spent 21.2 h on 7 focal-follows (mean: 3.0 h for each focal-follow, range: 1.9–4.6 h). During the focal-follows, a Barnett TS370 crossbow (Barnett Crossbows, Plano, IL, USA) was employed to deploy the Low-Impact Minimally Percutaneous Electronic Transmitter (LIMPET) satellite tag (SPLASH 10-373C, Wildlife Computers, Redmond, WA, USA) on the healthy adult whale [31,32].

We received time-series messages from Argos satellites passed overhead from the study area. Then, we filtered all original Argos-acquired locations by removing those repeated or erroneous locations in lower accuracy of location classes A or B [33]. We also excluded odd locations that appeared to be a single point and were not spatially clustered near the others [31,32,34]. We set a maximum whale's moving speed of 15 km/h for >1 h [8,32] and thus removed those locations with unacceptable moving speed between two consecutive points [34]. Furthermore, we used a Douglas Argos filter algorithm to calibrate all Argos-acquired locations [33,35]. At each filtered location, the water depth was measured from the bathymetric data layer downloaded from National Oceanic and Atmospheric Administration, USA (https://maps.ngdc.noaa.gov/; accessed on 2 July 2021). Between every two consecutive points, the linear distance and moving speed of the satellite-tracked whale were calculated [31,32,36]. Note that the linear distance, route, and moving speed estimated based on Argos locations should be interpreted and understood with caution, as the true movement shall be more complex than the estimated, and the linear distance and moving speed are usually underestimated or biased to some extent due to the location accuracy [35–37]. In this study, statistics are reported as mean ± standard deviation (SD), unless otherwise stated.

## 3. Results

Large-boat surveys were conducted in two survey days on 24 and 25 April 2021, while fixed-point observation and speedboat focal follows were implemented in all five survey days between 21 and 25 April 2021 (Table 1 and Figure 1C,D). The Bryde's whale could be observed on every survey day from 21 to 25 April (Table 1), but the occurrence locations were concentrated in a small area of ~20 km² (Figure 1C,D). The whale group size varied between 3 and 10 (Table 1). The mother–calf pair could be observed on every survey day (Table 1), and at least two pairs were photographically identified (Figure 2A–C). The tread-water feeding behavior was recorded for the groups on 21, 22, and 25 April (Table 1 and Figure 2D–F). On every survey day, whale-watching activities could be observed around Weizhou Island (Figure 2G), and the number of whale-watching boat trips per day ranged from 2 to 23 (Table 1).

**Table 1.** Summary of survey information on Bryde's whales around Weizhou Island, Beibu Gulf, China, from 21 April to 25 April 2021.

| Date | Large-Boat Survey | Fixed-Point Observation | Speedboat Focal-Follows | Group Size | No. of Mother–Calf Pair | Presence of Tread-Water Feeding | No. of Whale-Watching Trips |
|---|---|---|---|---|---|---|---|
| 21 April 2021 | No | Yes | Yes | 7–10 | 1 | Yes | 23 |
| 22 April 2021 | No | Yes | Yes | 7 | 1 | Yes | 16 |
| 23 April 2021 | No | Yes | Yes | 2–3 | 1 | No | 21 |
| 24 April 2021 | Yes | Yes | Yes | 2 | 1 | No | 5 |
| 25 April 2021 | Yes | Yes | Yes | 3 | 1 | Yes | 2 |

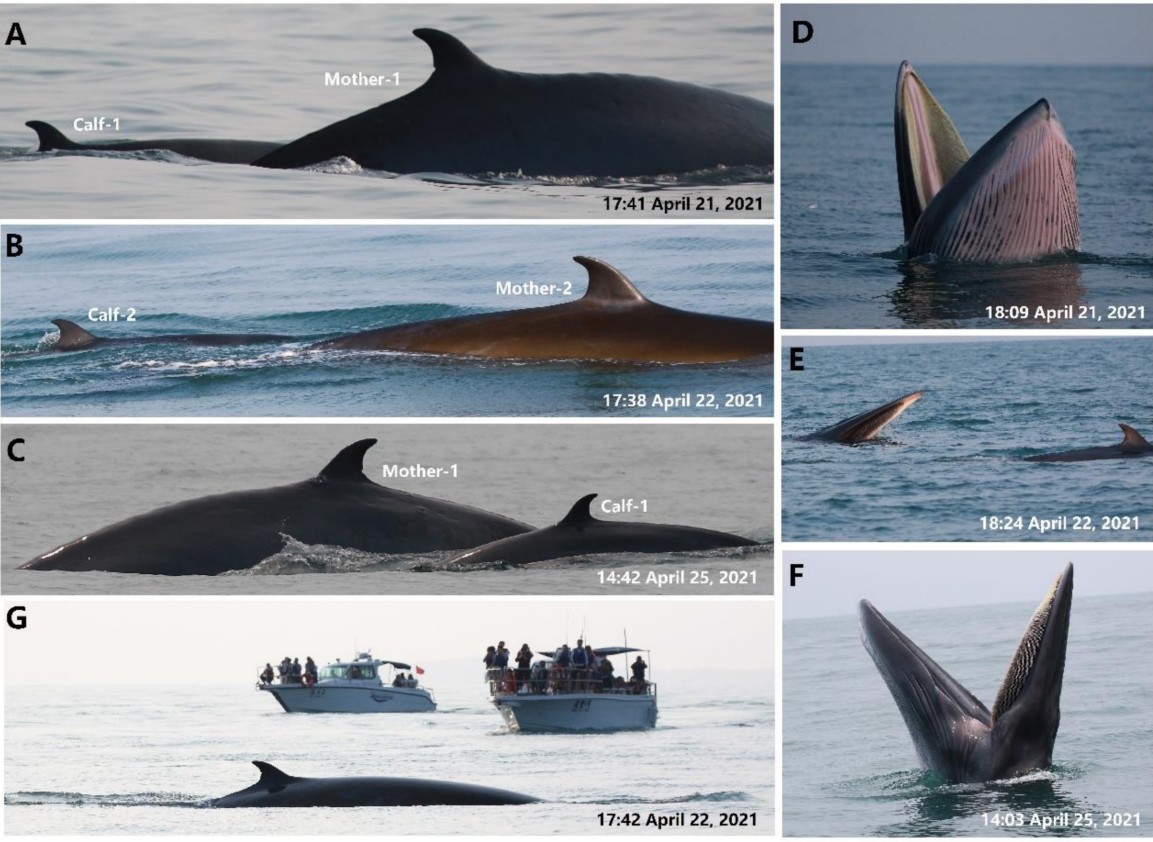

**Figure 2.** Photographic examples of (**A–C**) mother–calf pair of Bryde's whales in the Beibu Gulf on 21, 22, and 25 April 2021. Two mothers (Mother-1, -2) and two calves (Calf-1, -2) were identified; (**D–F**) tread-water feeding behaviors of Bryde's whales on 21, 22, and 25 April 2021; (**G**) whale-watching activities in the waters around Weizhou Island.

An adult Bryde's whale (BE-BG-ST01) was tagged on 25 April 2021 (Figure 3A,B), which has ever been photographically captured on 21 April 2021 (Figure 3C). It was subsequently satellite-tracked for 5 days 7 h 6 min (~6 days), spanning from 25 to 30 April (Figure 3D). In total, 149 Argos satellite locations were obtained (mean: 25 per day) (Table 2). After the process of filtering and calibration, 71 locations were retained (mean: 12 per day) (Table 2). At each satellite-tracked location, the water depth was measured as $42.1 \pm 24.8$ m on average, ranging from the shallowest $6.5 \pm 1.9$ m ($n = 11$) on 25 April to the deepest $86.3 \pm 5.4$ m ($n = 10$) on 30 April (Table 2 and Figure 3D). Throughout the satellite-tracking period, the whale's cumulative linear movement distance was estimated at 464 km, ranging from the shortest 14.7 km on 25 April to the longest 129.5 km on 29 April (Table 2 and Figure 3D). During the satellite-tracking period, the whale's moving speed was estimated at $5.33 \pm 4.01$ km/h, with the smallest daily speed of $2.31 \pm 3.52$ km/h on 26 April and the largest daily speed of $7.76 \pm 2.86$ km/h on 29 April (Table 2 and Figure 3E).

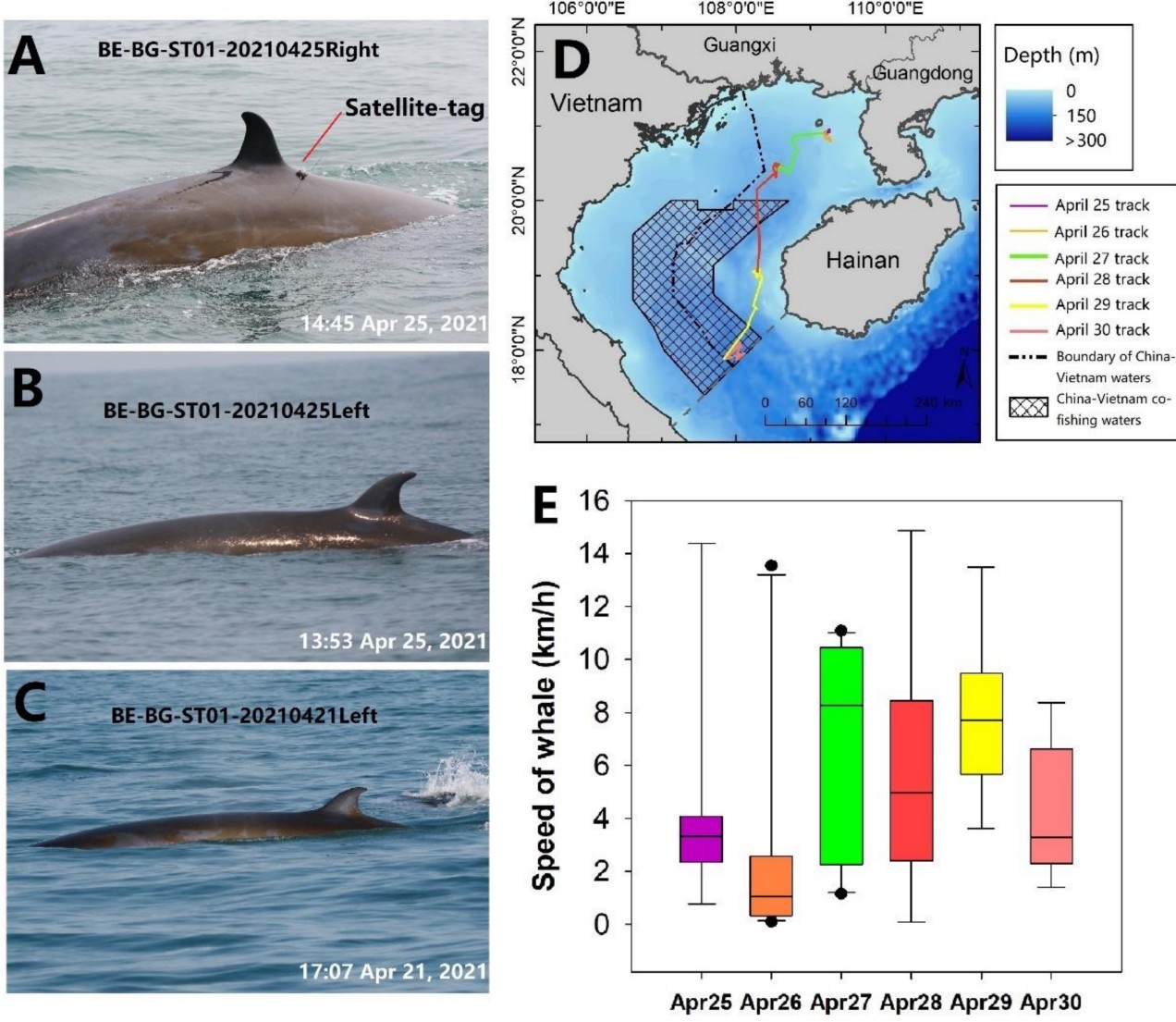

**Figure 3.** (**A**,**B**) Right and left lateral side of a satellite-tagged Bryde's whale (BE-BG-ST01) in the Beibu Gulf on 25 April 2021; (**C**) left side of BE-BG-ST01 on 21 April 2021; (**D**) daily tracking route and (**E**) moving speed of BE-BG-ST01 based on Argos-satellite locations in the Beibu Gulf from 25 to 30 April 2021.

**Table 2.** Information on a satellite-tracked Bryde's whale from 25 April to 30 April 2021.

| Date | No. of Argos Transmitted Locations | No. of Filtered Locations | Water Depth at Filtered Locations (m, Mean ± SD) | Linear Tracking Distance (km) | Moving Speed (km/h, Mean ± SD) |
|---|---|---|---|---|---|
| 25 April 2021 | 28 | 11 | 6.5 ± 1.9 ($n = 11$) | 14.7 | 2.90 ± 2.98 ($n = 11$) |
| 26 April 2021 | 25 | 11 | 12.5 ± 3.0 ($n = 11$) | 34.8 | 2.31 ± 3.52 ($n = 11$) |
| 27 April 2021 | 23 | 14 | 42.1 ± 8.7 ($n = 14$) | 68.7 | 6.83 ± 3.71 ($n = 14$) |
| 28 April 2021 | 25 | 13 | 51.8 ± 7.3 ($n = 13$) | 91.8 | 5.72 ± 4.54 ($n = 13$) |
| 29 April 2021 | 28 | 12 | 45.3 ± 19.8 ($n = 12$) | 129.5 | 7.76 ± 2.86 ($n = 12$) |
| 30 April 2021 | 19 | 10 | 86.3 ± 5.4 ($n = 10$) | 124.5 | 4.27 ± 2.45 ($n = 10$) |
| Average | 25 | 12 | 42.1 ± 24.8 ($n = 71$) | 77.3 | 5.33 ± 4.01 ($n = 71$) |
| Total | 148 | 71 | - | 464.0 | - |

## 4. Discussion

Scientific data on the occurrence and migration of baleen whales is essential to understanding their crucial distribution areas, which thus can facilitate further conservation management [28,38,39]. To the best of our knowledge, this is the first biologging attempt on baleen whales in Chinese waters, where cetaceans are poorly investigated but conservation is urgently needed [17,30]. The boat-based surveys, no matter in previous studies or in this study, only documented Bryde's whale encounters in a small area around Weizhou Island [26,27,29]. In this data-deficient context, it is extremely challenging to investigate potential occurrence areas and migration routes of Bryde's whales through boat-based surveys [29,30]. Alternatively, the biotelemetry technique can serve as an effective method to address this knowledge gap [4–7,31,32].

Our satellite-tracking data provided first insights into new occurrence areas of Bryde's whales in the Beibu Gulf, although only one individual was satellite-tagged with a relatively short-term tracking period. The whale's satellite-tracking route clearly showed that this species might move from a hotspot area to other areas that are less well-studied by boat-based surveys [26,27]. Our tracking data also indicated the likely migratory behavior of Bryde's whales in the Beibu Gulf, as the satellite-tracked whale was able to travel from the northern Beibu Gulf to the southern area. From 25 to 26 April 2021, the tracked whale moved at a low speed of ~3 km/h around Weizhou Island, resulting in a movement distance of <50 km per day. However, the tagged whale increased its daily moving speed to 4–7 km/h from 27 to 30 April, and it directionally traveled to the south with a minimum daily movement of 70–130 km. Overall, these findings reported the migratory behavior of Bryde's whales in the Beibu Gulf [26,27,29] and greatly expanded the known distribution areas of Bryde's whales in this gulf.

Our boat-based observations demonstrated the regular occurrence of Bryde's whales around Weizhou Island in a short survey period, which was highly in agreement with previous observations documented in this region [26,27,29]. Furthermore, the frequent presence of tread-water feeding behavior also confirmed that the waters around Weizhou Island were an important feeding ground for Bryde's whales in the Beibu Gulf, which has historically and currently been exploited as an important fishing ground [40,41]. Such a hotspot area of Bryde's whales might be strongly related to coastal upwelling systems and high primary productivity in the Beibu Gulf, providing whales with abundant food supplies such as pelagic–neritic anchovies, shads, herrings, and sardines [40–43]. More importantly, our boat-based observations firstly documented several mother–calf pairs for Bryde's whales in the Beibu Gulf, despite in a short survey period, suggesting that the waters around Weizhou Island might also be an important calving ground for Bryde's whales [39,44], if not within proximity to the study area.

However, it is still unclear whether there are any other functional areas (e.g., feeding, and calving grounds) for Bryde's whales in the Beibu Gulf and its adjacent waters. In recent decades, there have ever been recorded several occasional encounters and strandings of Bryde's whales along the coast of Hainan Island, China [17,18], Guangdong, China [17,45], and Vietnam [21]. These sporadic records indicated that this species might be distributed in a wider region, other than only in a small hotspot area. Furthermore, prey abundance and availability in time and space have been shown to affect the migratory behavior, seasonality, and abundance of Bryde's whales [39,44,46,47]. These studies provided additional evidence that foraging Bryde's whales are less likely to reside in one fixed place for a long period if their prey is on the move or not insufficient to supply for all individuals within a population [39,44,46,47]. Therefore, it is recommended that the continental shelf waters in the south of Hainan, middle and south of Vietnam should be investigated to explore the possibility of Bryde's whale occurrence in future research. In particular, more biologging attempts on Bryde's whales in the Beibu Gulf for long-term tracking could be a beneficial way [31,32,36,38].

In this study, the observed group size of Bryde's whales around the fixed-point observation area varied with the survey day, which was not surprising because the whales

might migrate from place to place, rather than residing around Weizhou Island. However, the group size on 21 April (7–10 individuals) and 22 April (7 individuals) was larger than the typical size with 1–2 individuals [8] and also was larger than the group size reported by previous studies in this region [26]. This is probably due to that the two large groups on those two days might be mixed groups such as transient feeding aggregations, which included several different social units [38,43].

Our satellite-tracking data provide baseline information on the Bryde's whale's moving speed. During the 6 tracking days, the whale's mean moving speed was estimated at 5.33 km/h on average, which was consistent with the typical speed of 2–7 km/h reported for this species [8] but was faster than those values documented for other baleen whale species, such as 1.3–5.4 km/h for the humpback whale *Megaptera novaeangliae* [32] or 2.5–4.5 km/h for the blue whale *B. musculus* [32,36]. Such difference might be due to the fact that the Bryde's whale is slimmer and smaller in body size than other rorquals, leading to the higher efficiency of swimming [32,48]. In addition, the higher speed estimates in this study might result from the migratory behavior from 27 to 30 April 2021. Thus, it is recommended to use suction-cup tags for more accurate estimates of swimming speed in future studies [28,37].

Unlike other baleen whale species, the Bryde's whale is not known to undertake long-distance migration, since only a few studies suggested that large-scale movement of Bryde's whales might reach up to 2000–3000 km [31,32,44]. Till now, there have been two confirmed hotspot areas of Bryde's whales in the South China Sea: one in the Beibu Gulf [26,27,29], and the other in the Gulf of Thailand [19,20,28]. The two gulfs have similar semienclosed environments with vast shallow waters and abundant fisheries resources [19,20,38,39], and the coastal distance between the two gulfs is less than 3000 km. Note that the highest encounter rates of Bryde's whales in the Beibu Gulf were mainly found between March and April [26,27], while the highest rates in the Gulf of Thailand were recorded between June and September [19,20]. Therefore, we recommend that the genetic relationship and possible connectivity of whales in these two gulfs should be further explored [12,49–51].

**Author Contributions:** Conceptualization, S.L., M.L. (Mingming Liu), and W.L.; methodology, M.L. (Mingming Liu), W.L., and M.L. (Mingli Lin); validation, S.L. and K.W.; resources, S.L.; data collection and curation, M.L. (Mingming Liu), W.L., B.L., and M.L. (Mingli Lin); writing—original draft preparation, M.L. (Mingming Liu), S.L., and W.L.; writing—review and editing, M.L. (Mingming Liu), W.L., M.L. (Mingli Lin), B.L., L.D. (Lijun Dong), P.Z., Z.Y., K.W., L.D. (Liang Dai), and S.L.; funding acquisition, S.L.; supervision: S.L. and L.D. (Liang Dai) All authors have read and agreed to the published version of the manuscript.

**Funding:** This research was financially supported by the National Natural Science Foundation of China (41422604), the "One Belt and One Road" Science and Technology Cooperation Special Program of the International Partnership Program of the Chinese Academy of Sciences (183446KYSB20200016), and the Key Deployment Project of Center for Ocean Mega-Science of the Chinese Academy of Sciences (COMS2020Q15).

**Institutional Review Board Statement:** The study was conducted according to the guidelines of the Law of the People's Republic of China on the Protection of Wildlife, and approved by the Department of Agriculture and Rural Affairs of Guangxi Zhuang Autonomous Region (Issued on 13 November 2020) and the Institute of Deep-sea Science and Engineering, Chinese Academy of Sciences with the Ethical Statement Number of IDSSE-SYLL-MMMBL-01 (Issued on 4 January 2013).

**Data Availability Statement:** Requests to the data presented in this article should be directed to SL (lish@idsse.ac.cn).

**Acknowledgments:** We are grateful to all the colleagues and students at the Marine Mammal and Marine Bioacoustics Laboratory. We would like to specially thank Jianlong Li, Hui Kang, and Lihua Liu for their participation in fieldwork. We also much appreciated Captain Jianwen Zhang and his crew for their assistance.

**Conflicts of Interest:** The authors declare no conflict of interest.

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
