# Peer review of "The First Attempt of Satellite Tracking on Occurrence and Migration of Bryde’s Whale (Balaenoptera edeni) in the Beibu Gulf"

_jmse, doi:10.3390/jmse9080796_

Round 1
Reviewer 1 Report
Liu et al describe data from a Bryde’s whale tagged with a satellite tag and report on the area through which it moved and its presumed average speed. Overall, the manuscript was easy to follow and the data was well-presented and builds nicely off prior research.
The only major comment I have is to more precisely consider the accuracy of ARGOS data. Line 120 does not make it clear which quality of ARGOS hits were used in the analysis. In my personal experience, quality A and B hits are highly inaccurate (could be 50 km off), and even quality 1 and 2 hits may be several km off. Additionally, the threshold of 25 km/h used to exclude successive hits is very high. While I believe Bryde’s whales have been observed at speeds close to that, there is no reason to suspect they would sustain those speeds for longer than a couple of minutes unless they were actively being chased. The inclusion of poor-quality hits and a threshold that is too high has likely skewed the reporting of average speed. Though it is unfortunately quite common in the literature to trust ARGOS locations as they are the best available data, it clearly can bias results. I highly recommend a more common-sense approach, for instance excluding quality B and A hits, as well as excluding points that appear to be “jumps”, by which I mean single locations that are not clustered near the others. I also recommend reducing your threshold, particularly if the gap between two locations is more than 15 minutes. For transiting whales swimming for more than two hours (i.e. points separated in space and in time by more than two hours), I would recommend a threshold closer to 10-15 km/hr (though hopefully you can find a better value from the literature than my guess here).
However, I understand that reporting all ARGOS positions is unfortunately common. If the authors wish to present all data they have collected, I recommend including some stronger caveats describing ways in which their speed estimates may be influenced by poor quality locations. I would also recommend including the whole data set (i.e. maps of each day with the locations listed or a supplemental data file) so that a reader could use their own judgment.
Otherwise the general description of the data is good and the paper will make a nice contribution to the literature. I only have additional minor comments, mostly highlighting minor ways in which the language could be improved.
Line 16- Clarify “was tracked” – by satellite, correct? A boat did not follow it for 6 days?
Line 23- “guarantee”- suggest “need”
Line 85- “we tracked the whale by biotelemetry technique”. This could mean many things, please clarify
Line 93- awkward sentence
Lines 108-109, each of these phrases needs the word “were”, e.g. “whether tread-water feeding activities were present”
Line 127- “were” should be “are” as current studies are present tense
Lines 153-158 When discussing maximums speeds and minimum speeds, do you mean the minimum daily average speed, or the minimum between two points?
Line 171 awkward sentence
Line 179- “where” should be “which”
Line 189 “where” should be “which”
Author Response
Thanks so much for your comments and suggestions. Thus, I resubmit the draft with a repsonse letter to your comments/suggestions one by one.
Please see the attachment.

Reviewer 2 Report
Overall I think this paper is in good shape and it's a nice addition that begins to fill a data gap. My comments are mostly minor. I have pointed out some typos and suggest spending some time in the Discussion making sure all of the writing is clear, as there are more confusing sentences in this section than in the Introduction, Methods, and Results, which are all quite clear.
Line 5: Extra comma at end of author list
Line 44: Binomial species names should be italicized. (See elsewhere in manuscript as well, e.g., line 49, line 62, 66, etc.)
Line 55: I think ‘non-existent’ is probably better than ‘non-existing’ here
Line 63: What do you mean by “Bryde’s whale demographics?” Are these distinct populations? Can you describe this in more detail?
Line 66-68: Can you clarify here that the assumption here is based on the habitat characteristics rather than observations of the whales themselves? That’s how I’m reading this sentence, but it’s a little unclear.
Line 70-71: Missing (D) in the caption
Line 80: ‘Except’ rather than ‘Exception’
Line 81-82: Do you mean scientifically described?
Line 91-92: For me, the HP and gross tonnage are less important from a methodological standpoint than metrics like maximum speed and range, but this is a minor detail and leaving it as it is would also be fine.
Line 94: Latitude and longitude should be reported with North first, then East (e.g., 20.93⁰N, 109.22⁰E)
Line 94: “north of” rather than “north off”
Line 109: “whether whale-watching boats were present”
Line 111-114: Could you add a sentence describing how many focal follows you conducted before you managed to attach a tag? Also how long? This is helpful for others planning tagging expeditions.
Line 144: I suggest adding an arrow to illustrate where the tag is in the photograph (Fig. 2F), as it can appear quite small in .pdf form.
Line 150: “retained” rather than “remained”
Line 154: “cumulative” rather than “accumulative”
Line 159-161: Figure 3. The text in the legend is really too small to read
Line 165: missing “the” before ‘occurrence’
Line 166: remove ‘the’ before ‘further conservation’
Line 171: What do you mean by “despite a wider surrounding area has been investigated?” Are you saying that the larger area has been surveyed and no whales were detected? Please clarify this sentence.
Line 171: I suggest “In this data-deficient context…”
Line 179: I suggest “…to other areas that are less well-studied by boat-based surveys”
Line 180-181: Are you suggesting migration between feeding grounds? In any case, I think you should add some discussion here about what you mean, as making a claim that you have documented migration based on 6 days of tracking data is perhaps more than the data can support – at least as it’s described here. I would recommend discussing this as ‘potential migration’ here, and adding some discussion about what else you would need to know in order to decide whether this was or was not real migration.
Line 219: Do you mean ‘temporary feeding aggregations?’
Line 230-232: Ah – so here you discuss why you think there is migration. I suggest re-organizing this section a little so that you can have a discussion about migration above, including this information about different speeds on different days.
Author Response

(The authors gave the same response as above.)
